# Electrochemotherapy as an Effective Alternative in the Treatment of Local Advanced Oral Squamous Cell Carcinoma: A Retrospective Analysis of Treated Cases

**DOI:** 10.3390/ijerph20065170

**Published:** 2023-03-15

**Authors:** Ida Barca, Francesco Ferragina, Elvis Kallaverja, Antonella Arrotta, Maria Giulia Cristofaro

**Affiliations:** 1Unit of Maxillofacial Surgery, Department of Experimental and Clinical Medicine, “Magna Graecia” University, Viale Europa, 88100 Catanzaro, Italy; 2Department of Medical and Surgical Sciences, “Magna Graecia” University, Viale Europa, 88100 Catanzaro, Italy

**Keywords:** electrochemotherapy, oral cancer, head and neck cancer, advanced squamous cell carcinoma, maxillofacial surgery

## Abstract

Advanced oral squamous cell carcinomas represent a major challenge for maxillofacial surgeons, oncologists and radiation therapists. They also account for a large share of healthcare costs. They respond little and/or poorly to conventional therapies (surgery, radiotherapy and chemotherapy). Electrochemotherapy is a new method used as a palliative treatment in patients with advanced cancer of the neck/head region who are not eligible for standard therapies. It combines the use of cytotoxic drugs with the physical principle of electroporation; it effectively controls the tumour locally and preserves organ function. To date, ECT has been little used for oral mucosal tumours, as this is difficult to access for electrodes. We report six cases of advanced oral squamous cell carcinoma treated with electrochemotherapy. This study aims to assess the debulking effect of cancer via ECT in patients with advanced oral squamous cell carcinoma. It also aims to assess the safety and tolerability of this treatment.

## 1. Introduction

Among all cancers of the oral and maxillofacial region, oral cancer (oral cancer OC) is certainly the most frequent: It is the eleventh most common tumour in the world [1]. More than 90–95% of tumours in this area are squamous cell carcinomas; the remaining 5–10% include adenocarcinomas, melanomas, sarcomas and lymphomas [1,2,3]. Advanced oral squamous cell carcinoma (AOSCC) still represents a global health problem due to its morbidity and mortality. Maxillofacial surgery plays a key role in the treatment of these tumours. Surgical excision with appropriate safety margins is considered the first-line treatment. Adjuvant therapy, both systemic (chemotherapy, immunotherapy and cancer growth inhibitors) methods and radiotherapy (RT), aims to reduce the risk of relapse. Although the complete removal of the tumour is the primary goal, the preservation of organ functions and a satisfactory quality of life are equally important. Functional disorders related to the disease and induced by treatments (especially speech and swallowing), aesthetic disfigurement, metastases and secondary tumours that arise in previously treated regions pose a challenge for doctors and surgeons dealing with this disease [3]. However, not all patients are suitable for these standard therapies (surgery, radiotherapy and systemic therapy) as in the case of locally advanced disease or important comorbidities. Precisely in these cases, alternative methods must be used. Despite introducing new systemic drugs and improving standard therapies, the results of existing treatments are not yet satisfactory. The search for new therapies to cure and/or combat these tumours’ growth and reduce discomfort, especially in elderly people who cannot undergo standard or advanced therapies for the disease, is very important. Electrochemotherapy (ECT) is a new therapeutic method that combines engineering technologies with medical and biological knowledge.

The term ECT was first used in 1991 by Mir et al. in a study describing the effect of local applications of cytotoxic electrical impulses on subcutaneous tumours after the intramuscular injection of bleomycin [4].

ECT is a topical ablative treatment based on the physical mechanism of electroporation (EP). EP involves the application of a pulsed electrical stimulus at high voltage and short intensity on cancer cells. This external electric field induces a temporary increase in the permeability of plasma membranes. This facilitates the cellular absorption of hydrophilic drug molecules, both by diffusion and by active transport. The drugs thus enter and accumulate in the cell’s cytoplasm, significantly increasing their cytotoxicity [5]. The most used anticancer drugs in ECT are bleomycin (a hydrophilic antibiotic with endonuclease activity) and cisplatin. They form adducts with DNA and lead to cell death, especially for apoptosis; preclinical studies have shown that anticancer activity increased by 8000 times for bleomycin and 80 times for cisplatin. This implies three important advantages: (1) a significant anticancer effect can be achieved with a minimum dose; (2) there is a strong reduction in systemic side effects; (3) anti-vascular effects (marked in well vascularised tumours) due to direct vasoconstriction and vascular disruption following the endothelial death of the tumour vessels. In addition, the destruction of cancer cells triggers a local immune response that can contribute to the overall effectiveness of ECT [6,7].

Many authors have shown that the window of time for the application of ECT in head and neck cancers is 8–28 min after bleomycin administration [8]. Chemotherapy drugs can be administered in three different ways: intravenous, intraarterial and intratumour. The intratumour injection is the best method for poorly vascularised lesions; intravenous drug administration is more convenient in the case of multiple tumours or extensive lesions [9]. The operating procedures were developed and validated in 2006 by the European Standard Operating Procedures for Electrochemotherapy (ESOPE) study, which accurately described the dosage, times and value of the electric field and the assessment of treatment response for various diseases [10]. Thanks to these guidelines, ECT has mainly been used to date in the treatment of skin cancer: basal cell and squamous cell carcinomas, melanoma skin metastases, Merkel cell carcinoma and others [11,12,13]. In the mucosal tumours of the head and neck region, experience with ECT is limited, mainly due to the anatomical complexity of the region and poor accessibility to tumours [14].

## 2. Materials and Methods

### 2.1. Aim

This study aims to assess the neoadjuvant effect of ECT in patients with advanced oral squamous cell carcinoma (AOSCC) who are not eligible for surgery or who have rejected it because of its strong disfigurement (both from the functional and aesthetic points of view).

### 2.2. Endpoints

The primary endpoint is to verify tumour debulking after ECT treatment. The secondary endpoint is to assess the safety and tolerability of ECT treatment.

### 2.3. Sample and Setting

This prospective study includes patients with advanced oral squamous cell carcinoma admitted to the Department of Maxillofacial Surgery of the University Hospital “Magna Graecia”, Catanzaro, Italy. Each patient was evaluated and treated by an expert team from 1 January 2020 to 31 January 2022.

The study was carried out according to the guidelines set out in the Helsinki Declaration and was approved by the Ethics Committee of the “Magna Grecia” University, Catanzaro, Italy (protocol number 003589_15).

### 2.4. Inclusion and Exclusion Criteria

The inclusion criteria of the study are age ≥ 18 years; progressive disease (single recurrence on T and N0) of the oral cavity; absence of other systemic metastases in any site outside the locoregional recurrence; histological diagnosis of squamous cell carcinoma; performance status with Karnofsky ≥ 70 and/or WHO ≤ 2; life expectancy > 3 months; ability to understand the information given and signed informed consent; and ineligibility for standard treatment.

The exclusion criteria are other symptomatic lesions not under control, lesions not suitable for ECT (bone invasion, large vessel infiltration, etc.), known adverse reactions to bleomycin and chronic renal dysfunction.

### 2.5. Procedure

Before any treatment, all patients signed an informed consent form to participate in the study. A researcher clarified the details of the study extensively and explained that their participation was voluntary and that they could not be identified. It was explained that each participant/patient could leave the study at any time. In addition, potential participants had ample opportunity to ask questions.

Each patient was examined at the maxillofacial surgery department: An experienced team collected medical records and carried out a careful objective examination. The patient’s medical history was documented, paying attention to the therapies taken and the possible oncological history (previous tumours, therapies administered and response to the latter). Risk factors such as cigarette smoking and alcohol intake were investigated. As for smoking, we divided the patients into heavy smokers (patients who smoked ≥ 20 cigarettes a day), moderate smokers (patients who smoked < 20 cigarettes a day) and non-smokers. As for alcohol consumption, we divided the patients according to the intake of alcohol units per day (each alcoholic unit corresponds to 12 g of ethanol): strong drinker (intake of ≥3 units per day in the last month), moderate drinker (intake of ≤2 alcohol units per day in the last month) and non-drinkers. The patient underwent a comprehensive physical examination that included weight, height and vital signs.

A biopsy was performed under local anaesthesia. Sampling was performed at the level of the most suggestive macroscopic appearance portion of malignancy. The sample was then evaluated histologically. For grading, Broder’s classification was used which considers the degree of differentiation of the tumour. According to how far the neoplasm deviates, in its histological aspect, the following has been identified from the normal tissue: G1 (low grade) well differentiated tumours (<25% of undifferentiated cells); G2 (intermediate grade) moderately differentiated tumours (<50% of undifferentiated cells); G3 (high grade) poorly differentiated tumours or undifferentiated tumours (>50% of undifferentiated cells).

Each patient performed total body computed tomography (CT) and magnetic resonance imaging (MRI) of the head and neck region with a contrast medium. A total body PET/CT with the administration of ^18^F-FDG (positron emission tomography/computed tomography with the administration of ^18^fluoro-desoxyglucose) was performed only if deemed necessary.

Upon hospital admission, all patients were subjected to a pre-anaesthesia assessment following the procedures of good medical practice.

The tumour was measured and then staged. A multidisciplinary team of maxillofacial surgeons, oncologists, radiotherapists and radiologists developed and evaluated an indication for standard therapies or alternative therapies such as ECT. ECT was performed under general anaesthesia to reduce pain and muscle spasms induced by the technique, according to the ESOPE protocol. The ECT treatment was performed in an operating room in sterility. The anticancer drug used was bleomycin; it was prepared in the hospital’s pharmacy using standard procedures for this drug. The bleomycin dosage was adjusted according to standardised oncology practice. In this manner, the concentration of bleomycin in the interstitial tissue was enough to kill all cancer cells that divide, completely sparing normal non-dividing cells.

During the first minute after anaesthesia induction, bleomycin bolus was given intravenously. The ECT procedure started 8 min after the intravenous bolus injection of the anticancer drug, to ensure complete spreading to all cancer cells. The electrode type used to administer the electrical impulses was the “finger” probe (Figure 1).

Electrical impulses were generated via Cliniporator^TM^ (IGEA SpA, Carpi, Italy). Both the electrodes and the pulse generator were CE certificated. The electrode was inserted both inside and around the tumour. The delivery parameters were 8–96 pulses of 140–1000 V, 100 µs duration and at 5000 Hz repetition frequency. The number of electrical impulses provided was calculated based on the size of the neoplasm (to cover the entire tumour field). In addition, after the administration of each pulse, the validity of the administration was checked on the monitor.

After finishing the ECT treatment, the patient was woken up and transferred to the maxillofacial surgery department. The patient was thus monitored and any toxicity and/or adverse events due to bleomycin were managed. Post-operative analgesia was prescribed following the usual procedures of good medical practice. The duration of the ECT post-treatment hospitalisation was 2 days (increased if the patient’s condition required it).

### 2.6. Response to Treatment

Responses to the ECT treatment were assessed as follows: (a) clinical checks were carried out at 1, 3, 6, 9 and 12 months from ECT; (b) radiological examinations (CT or MRI) were performed at 3 and 12 months from ECT; (c) neck lymph node ultrasound was performed at about 6 months from ECT; (d) about three months after ECT, all patients underwent multiple biopsies of the treated area. Photographs of the lesions were taken (in such a way that the patient’s distinctive features were not captured) at each preoperative and postoperative clinical check-up and during surgery (for scientific purposes only). For each patient, the following parameters were evaluated: local cancer control, pain control and quality of life.

The RECIST Criteria (Response Evaluation Criteria in Solid Tumors) were used by radiologists for the local assessment of cancer. In addition, all lesions were documented with photos to evaluate treatment results in terms of reducing the local extent of the disease.

Patients recorded pain according to the Visual Analogue Scale (VAS). The subjects enrolled in the study received precise and detailed instructions to write a daily diary for a month after the ECT treatment. Patients were asked to draw a sign representing the level of pain experienced on the VAS line and to record the number of painkillers taken as well as any side effects, especially bleeding and impaired swallowing and respiratory function.

The term health-related quality of life (HRQOL) is often described as follows: “A term referring to the health aspects of quality of life, generally considered to reflect the impact of disease and treatment on disability and daily functioning. However, more specifically HRQOL is a measure of the value assigned to the duration of life as modified by impairments, functional states, perceptions, and opportunities, as influenced by disease, injury, treatment, and policy” [15]. Life-quality tests are the basis for assessing the condition of oncological patients. Tests to evaluate HRQOL were provided for patients before treatment and at each follow-up visit. EORTC QLQ-C30 (version 3) and EORTC QLQ-H&N43 were the questionnaires used.

### 2.7. Statistical Analysis

Both descriptive and regressive statistical analyses were performed on the recorded data. Descriptive statistical analysis was performed using categorical data’s central tendency indices and absolute and relative frequencies. Regressive statistical analysis was performed using two variable correlation tests, Student’s *t*-test and analysis of variance (ANOVA). Both were performed using the GraphPad program (GraphPad Company, San Diego, CA, USA). Statistical significance was accepted at *p* < 0.05. If the percentages do not add up to 100, this is because the data have been rounded to the nearest 1%

## 3. Results

A total of 13 patients (7 males and 6 females) were evaluated for AOSCC, and only 6 (3 males and 3 females) fit the inclusion criteria and were included in the study. The average age of the patients was 72.5 years with a range of 54–85. The analysis of life habit data showed that all enrolled patients were smokers (five strong smokers and one moderate smoker); as for alcohol consumption, one patient was a nondrinker (one woman) and five were drinkers (two men were strong drinkers; one man and two women were moderate drinkers). Patient characteristics and the number of ECT sessions performed are shown in Table 1.

Three had tumours of the palate (two males and one female), two had tumours of the buccal mucosa (one male and one female) and one had tumours of the oral floor (one female). None of the patients were eligible for standard therapies.

In accordance with TNM staging, all patients were staged as T3; only one was positive for lymph node metastases (a woman with AOSCC-G3 of the oral floor).

According to grading, four tumours were classified as moderately differentiated G2 (three at the level of the palate and one at the level of the buccal mucosa) and two tumours were classified as poorly differentiated G3 (one at the level of the buccal mucosa and one at the level of the oral floor).

In two cases (33.33%), complete remission of the lesion was achieved after a single ECT session: in particular a male with AOSCC-G2 of the palate (Figure 2A,B) and a female with AOSCC-G2 of the buccal mucosa.

In one case (16.66%), specifically a male with AOSCC-G2 of the palate, complete remission of the lesion was achieved after two ECT sessions. In two cases (33.33%), specifically a male with AOSCC-G3 of the buccal mucosa (Figure 3A,B) and a female with AOSCC-G2 of the palate, complete remission of the lesion was achieved after three ECT sessions.

The disease’s remission was confirmed with a biopsy three months after ECT. Only one (16.67%) patient, specifically a female with AOSCC-G3 of the oral floor, experienced tumour regression after a single ECT session but refused further treatment. All patients showed a drastic reduction in VAS scores from immediate post-treatment: from an average of 8.75 and 8.18 on the first and third days after ECT up to an average of 5.4 fifteen days after ECT and 2.37 one month after treatment. The VAS scores reported by patients are represented in Table 2.

VAS scores are significantly reduced from the first day to the fifteenth and thirtieth day. Despite this, a Student’s *t*-test analysis showed no statistical significance, exhibiting a *p*-value of 0.1664. The ANOVA test was also not statistically significant (*p*-value = 0.1378).

Tests for HRQOL have also shown an improvement in the quality of life with recovery in daily nutrition and activities. No major complications, such as impaired swallowing and/or respiratory function, occurred in any of the treated cases that required urgent handling.

## 4. Discussion

OSCC is a frequent tumour, with variable prognosis depending on the disease’s location, classification and spread. Early-staged OCSCs have a favourable prognosis and are usually treated with surgery or RT. Although more and more advanced therapeutic strategies have been developed over the years, 5-year survival rates are approximately 50% [16,17,18], with most of them being diagnosed at an advanced stage. The treatment of these patients can be difficult, especially if they have previously been treated with RT and/or surgery. In addition, many patients are of advanced age and have a low performance status, have numerous comorbidities and have locally advanced tumours with the involvement of important anatomical structures. They are, therefore, not suitable for surgery, radiotherapy and/or systemic therapy. For all these reasons, in recent years there has been a growing interest in therapies such as ECT. ECT is able to control the tumour locally and, at the same time, preserve organ function [19]. This new method combines the use of cytotoxic drugs with the application of a short-intensity high-voltage pulsed electrical stimulus. There are two main effects on cancer: (1) a direct effect on tumour cells of transient change in plasma membrane permeability; (2) an anti-vascular effect, directed against the tumour vessel [20,21]. The first effect allows the transport (diffusion or active transport) of the cytotoxic drug within the cytoplasm of the cancer cell. In this way, very low doses of the drug cause a very powerful effect: bleomycin cytotoxicity increases from 300 to 700 times [20,21]. The second effect is due to the direct vasoconstriction induced by the electric impulse and is very pronounced in well vascularised tumours. In this way, vascularisation is interrupted by the necrosis of the endothelial cells of the cancer vessels.

This study used Cliniporator^TM^ (IGEA S.p.A., Carpi, Italy) to induce electroporation by applying electrical impulses. The Standard Operating Procedures were established as a result of a European Commission-funded project under the Fifth Framework Program [6,10,22,23,24,25]. Our study confirmed its clinical efficacy with an acceptable safety profile. All patients were treated under general anaesthesia to reduce patient discomfort. The application of electrical impulses determines the appearance of muscle spasms, which are very painful and can be psychologically traumatic [26,27]. Our study demonstrates that ECT plays a significant role in symptom control (especially pain) and the management of advanced oral mucosa tumours. It also played an important role in limiting disease progression and debulking it. It represents the most effective therapeutic option because it is better tolerated and accepted by patients. ECT-related toxicity has not been observed. All patients experienced a reduction in tumour-related symptoms within two weeks of treatment. In addition, they reported an improvement in their quality of life with the resumption of normal oral nutrition and daily activities. Even in the case of partial responses to treatment, it induced an improvement in symptomatology (especially pain and bleeding) and a reduction in hospitalisation length (and, therefore, in public health expenditure). In addition, it could have been repeated without worsening the patient’s life quality, but rather would improve their symptoms. To date, very few cases of AOSCC treated with this technique are documented in the literature. The objective response rates reported in the literature are very promising (ranging from 56% to 100%) and the data reported in our study underline this trend, observing an objective response rate of 83.33%. ECT, in the reported cases, emerges as a safe and well tolerated method and exhibits no side effects related to a drug or pulses. This figure is probably the best indicator of the applicability of ECT: patients agree to be treated, even for more than one session. Although effective, the technique certainly needs to be improved by the technical devices used. The areas reached by the electrode and effectively treated are limited, and the electrodes should be more manageable [28]. Future studies are therefore necessary to confirm the effectiveness and tolerability of this method, both as a palliative and as a cure.

## 5. Conclusions

The effectiveness of ECT is influenced only by the size of the tumour and not by the histological nature of lesions. It is still considered a palliative treatment reserved for patients in the advanced stages of the disease. Our prospective study demonstrates the long-term effectiveness of ECT in the treatment of advanced oral squamous cell carcinomas. The primary endpoint of the study (to verify the debulking of the tumour after ECT treatment) was reached, obtaining an objective response rate of 83.33% (evaluated by RECIST criteria). A partial response was obtained only in one case (16.67%): disease control and reduction in painful symptoms after a single ECT session. The overall results of the treatment performed on the six patients in the study show good subjective tolerance of ECT in all cases (100%) in the absence of significant adverse effects. In this way, the secondary endpoint was reached and demonstrated (assessing the safety and tolerability of treatment).

ECT allows improved preservation of tissues, fewer side effects, shorter hospitalisations and greater repeatability of treatment (both individually and combined with other methods), resulting in lower costs. The main limitation of this study is certainly the small sample. However, this attests to the capacity of standard therapies in AOSCC management. For this reason, few cases of AOSCC do not respond to standard therapies. Another important limitation is the impossibility of treating all types of cancer. Electrodes can only reach the submucosal layer and in some cases the muscle. The development and research in the field of electrode design, image-driven and robotic-assisted approaches, kinetics of bleomycin, and interactions with the immune system give hope that ECT will play a more decisive role in these tumours in the future.

## Figures and Tables

**Figure 1 ijerph-20-05170-f001:**
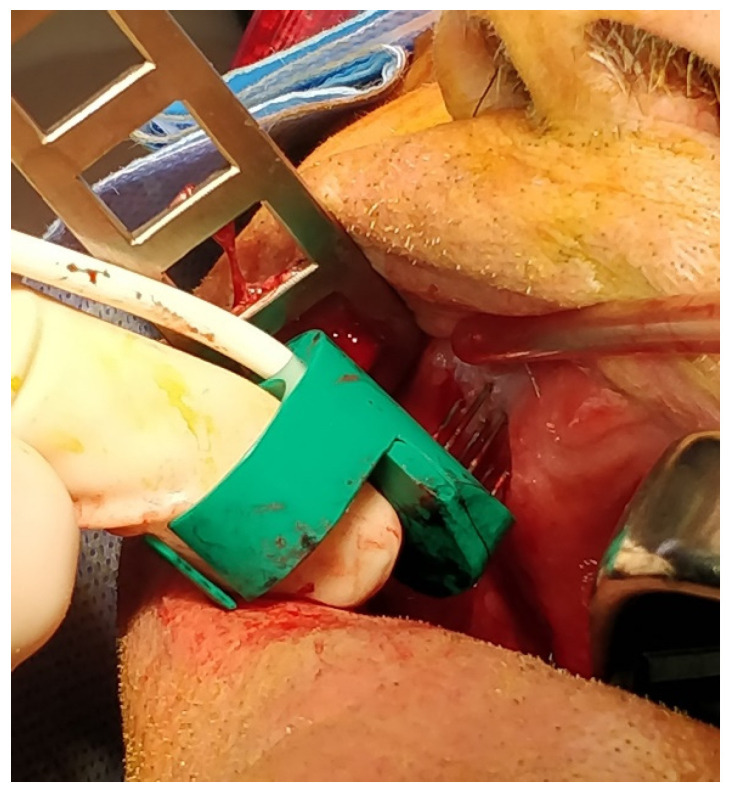
“Finger” electrode used for the delivery of electrical impulses.

**Figure 2 ijerph-20-05170-f002:**
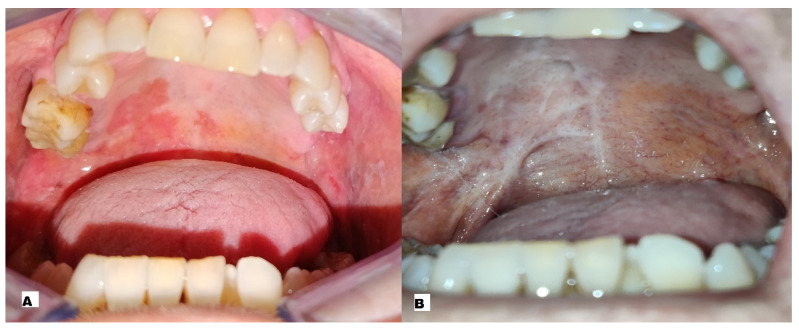
(**A**) Pre-treatment image showing oral squamous cell carcinoma such as an erythematous non-bleeding de-epithelialised area with jagged margins, of the palate; (**B**) post-treatment image showing scarring from a biopsy and ECT with no sign of malignancy.

**Figure 3 ijerph-20-05170-f003:**
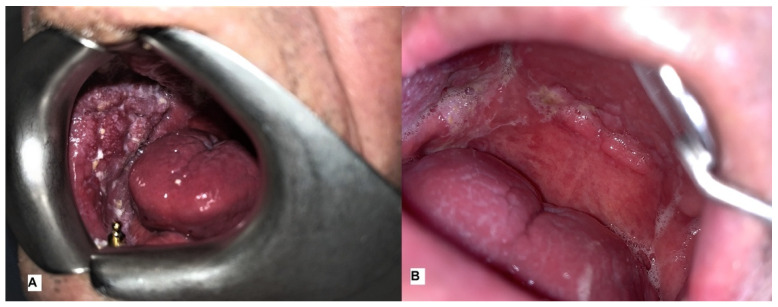
(**A**) Pre-treatment image showing an erythroleucoplase exophytic non-bleeding area with jagged margins of the buccal mucosa; (**B**) post-treatment image showing scarring from a biopsy and ECT with no sign of malignancy.

**Table 1 ijerph-20-05170-t001:** Characteristics of the patients and tumours.

Patient	Sex	Age	Smoker	Alcohol	Grading	Localisation	Stage	N° of Sessions of ECT
1	M	85	Strong smokers	Moderate drinker	G2	Palate	T3N0	1 ECT session
2	M	68	Strong smokers	Strong drinker	G2	Palate	TEN0	2 ECT sessions
3	M	71	Strong smokers	Strong drinker	G2	Buccal mucosa	T3N0	1 ECT session
4	F	54	Strong smokers	Non-drinker	G2	Palate	T3N0	3 ECT sessions
5	F	76	Moderate smoker	Moderate drinker	G3	Buccal Mucosa	T3N0	3 ECT sessions
6	F	81	Strong smokers	Moderate drinker	G3	Oral Floor	T3N1	1 ECT session

**Table 2 ijerph-20-05170-t002:** Patients’ single and cumulative VAS scores regarding pain after ECT treatment.

	VAS Score after 1 Day from ECT Treatment	VAS Score after 2 Days from ECT Treatment	VAS Score after 15 Days from ECT Treatment	VAS Score after 30 Days from ECT Treatment
patient 1	9	9	5, 6	3, 6
patient 2	8, 6	8, 2	6	2, 1
patient 3	8, 6	7, 8	5, 3	2, 6
patient 4	7, 9	7, 6	4, 9	0, 9
patient 5	9, 3	8, 1	5, 6	1, 8
patient 6	9, 1	8, 4	5	3, 2
Average VAS Score	8.75	8.18	5.4	2.36

VAS: Visual Analogue Scale; ECT: Electrochemotherapy.

## Data Availability

The data presented in this study are available upon request from the corresponding author.

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
