# Peer review of "Electrochemotherapy as an Effective Alternative in the Treatment of Local Advanced Oral Squamous Cell Carcinoma: A Retrospective Analysis of Treated Cases"

_ijerph, 2023, doi:10.3390/ijerph20065170_

Round 1

Reviewer 1 Report

The manuscript to the paper “Electrochemotherapy as an Effective Alternative in the Treatment of Local Advanced Oral Squamous Cell Carcinoma: A Retrospective Analysis of Treated Cases” is well written and has an interesting topic.

This is a prospective study, and often requires more labour and takes longer time than retrospective ones. The quality is often better in prospective studies as clinical and lifestyle information is often more comprehensive. This is also true for this study.

Location nicely described for each patient, and also VAS.

Some minor comments:

Material and methods; Perhaps too many options for differentiation – 3 would have been enough (page 3, line 140-143).

Shift between past tense and presens (page 4, line 158, 162, 171). Would have preferred all in past tense.

In general, one could be concerned that this treatment would only be effective with superficial tumours, but all were T3. The pictures illustrate very nicely the healing of the tumour areas, but one could be concerned that there would be malignant cells in the deeper part of the tissue causing residual growth of tumour tissue. This is partly, and quite nicely discussed in the discussing/conclusive part of the paper.

Author Response

On behalf of all authors, we thank the reviewer for his prompt and timely reply. Thank you for the many compliments placed on our work and for the valuable suggestions. All required changes have been made.

Reviewer 2 Report

Barca et al performed a smaller study with 6 enrolled patients aiming to evaluate the effect of debulking the locally advanced oral squamous cell carcinoma by use of electrochemotherapy as well as to assess the safety and tolerability of this treatment.          

The manuscript is relatively relevant for the field, is well structured but is not written clearly.

Most cited references are recent. Some of them are missing (in correlation with the comments added in the attached document). There is no excessive number of self-citations.

Manuscript is moderately scientifically sound; it is of descriptive nature.

There are quite a few weaknesses of the paper. The main weakness of the study lies in the fact that only 6 patients have been enrolled. Authors should better explain the difference between advanced and locally advanced oral SCC. Quite a problem is in the fact that Figures are not very persuasive, especially Figure 2A in which no tumor is present, only demarcated, erythematous area could be seen (while study is intended to deal with advanced oral squamous cell carcinomas; this one could respond to in situ carcinoma as well). It would be great if authors could replace Figure 2 for some more convincing one. Figure 3 is not ideal, as well. At least 3B should be rotate in the right direction for 90°for easier and better comparison with Fig 3A. Since authors write also about RECIST criteria for the tumor response, some radiological images could be added for comparison between naive, non-treated tumors and images after the ECT procedure.

Throughout the paper there are many parts that are not well and clearly written and should be rephrased and /or re-written for better understanding and improvement the quality of the paper. At a few points authors give too extensive descriptions instead of giving scientific facts supported by appropriate citations.  A word document with many comments is attached. There are too many corrections and comments to list them one by one.

Extensive editing of English language and style is required. It seems that authors used some of the on-line translating services (many expressions are inadequate and should be replaced by more appropriate ones).

All references should be written in the same manner.

Pay attention to Conflicts of Interest statement: Authors declare that they have no conflict of interest. No research involving humans and/or animals was performed. A second sentence (part) should be removed from the conflicts of interest statement into the ethical part. As well, it should be corrected since study involved human subjects.

Author Response

On behalf of all authors, we thank the reviewer for his prompt and timely reply. All required changes have been made:

  • Only six patients are considered because most patients with AOSCC, fortunately, have been treated with standard therapies.
  • The figures and related Captions have been changed, as suggested by the reviewer.
  • Figure 2A shows an erythematous area in which a biopsy was performed: histology attested a squamous cell carcinoma infiltrating the underlying structures (not in situ carcinoma).
  • As suggested, many parts of the text have been replaced or rewritten. it has also been used the suggested online translating services.
  • The references have all been checked and standardized.
  • The "Conflicts of Interest statement" section has been modified.

Thank you for the valuable suggestions.

Round 2

Reviewer 2 Report

None, already done